# DELTA50: A Highly Accurate Database of Experimental ^1^H and ^13^C NMR Chemical Shifts Applied to DFT Benchmarking

**DOI:** 10.3390/molecules28062449

**Published:** 2023-03-07

**Authors:** Ryan D. Cohen, Jared S. Wood, Yu-Hong Lam, Alexei V. Buevich, Edward C. Sherer, Mikhail Reibarkh, R. Thomas Williamson, Gary E. Martin

**Affiliations:** 1Analytical Research and Development, Merck & Co., Inc., Rahway, NJ 07065, USA; 2Department of Chemistry and Biochemistry, Seton Hall University, South Orange, NJ 07079, USA; 3Department of Chemistry and Biochemistry, University of North Carolina Wilmington, Wilmington, NC 28409, USA; 4Department of Computational and Structural Chemistry, Merck & Co., Inc., Rahway, NJ 07065, USA

**Keywords:** NMR, DFT, chemical shift predictions, benchmark, computational chemistry

## Abstract

Density functional theory (DFT) benchmark studies of ^1^H and ^13^C NMR chemical shifts often yield differing conclusions, likely due to non-optimal test molecules and non-standardized data acquisition. To address this issue, we carefully selected and measured ^1^H and ^13^C NMR chemical shifts for 50 structurally diverse small organic molecules containing atoms from only the first two rows of the periodic table. Our NMR dataset, DELTA50, was used to calculate linear scaling factors and to evaluate the accuracy of 73 density functionals, 40 basis sets, 3 solvent models, and 3 gauge-referencing schemes. The best performing DFT methodologies for ^1^H and ^13^C NMR chemical shift predictions were WP04/6-311++G(2d,p) and ωB97X-D/def2-SVP, respectively, when combined with the polarizable continuum solvent model (PCM) and gauge-independent atomic orbital (GIAO) method. Geometries should be optimized at the B3LYP-D3/6-311G(d,p) level including the PCM solvent model for the best accuracy. Predictions of 20 organic compounds and natural products from a separate probe set had root-mean-square deviations (RMSD) of 0.07 to 0.19 for ^1^H and 0.5 to 2.9 for ^13^C. Maximum deviations were less than 0.5 and 6.5 ppm for ^1^H and ^13^C, respectively.

## 1. Introduction

Prediction of NMR chemical shifts using computational quantum chemistry, including density functional theory (DFT), is a well-established methodology that can significantly decrease the likelihood of structure determination errors [1,2,3,4,5,6,7,8]. Such calculations can aid in NMR peak assignments [9,10], as highlighted in our recent example of misassigned beta-lactam carbonyl chemical shifts [11]. They have been used to confirm organic [12,13,14,15], inorganic [16,17], and organometallic [18,19] reaction products, particularly those with unexpected or unusual molecular structures [20,21,22], and they have been applied in complex speciation studies [23,24,25]. Perhaps most notably, chemical shift calculations were used to revise incorrectly reported natural product structures: aquatolide [26], vannusal B [27], glabramycin C [28], and hexacyclinol [29]. For elucidation of spectroscopically challenging natural products, Buevich and Elyashberg demonstrated that DFT NMR calculations could enhance the performance of computer-assisted structure elucidation (CASE) [30,31,32]. More recently, such calculations have been used to determine the conformations of cyclic peptides in solution [33,34], and they have been applied to biomolecules, such as nucleic acids [35,36,37], carbohydrates [38,39,40], and proteins [41,42,43]. Calculations of chemical shifts, via shielding tensors, have also been combined with solid-state NMR to help refine X-ray diffraction data of proteins [44] and to determine the packing arrangements of microcrystalline material [45], a practice referred to as NMR crystallography [46].

DFT is among the most widely used computational chemistry tools for chemical shift calculations due to the combination of accuracy and efficiency [3,7,47]. Application of DFT for structure problem solution involves judicious choice of both a density functional and basis set, which can be obtained from benchmark studies on a test set of compounds, often from 20 to 100 small- to medium-sized, rigid molecules that have readily available experimental data. A multitude of DFT benchmark studies have been reported in the literature (Table 1), yet these studies have presented conflicting conclusions regarding the most accurate method. For instance, Konstantinov and Broadbelt found BMK to be the best performing functional for chemical shift predictions [48], while Toomsalu and Burk found it to be the worst performing [49]. Granted, these two studies used different solvent conditions (viz., toluene-*d*_8_ and CDCl_3_), but even for the same solvent system there have been significantly different findings, such as Benassi recommending WP04 for δ_C_ predictions in CDCl_3_ based on a benchmark test set of 104 diverse, small organic molecules [50], while Buß and Koch found the performance of WP04, for a test set of 24 heterocycles in CDCl_3_, to be the second worst performing density functional for δ_C_ [51]. Paradoxically, the worst performing functional, WC04, from Buß and Koch’s study [51] was specifically parameterized for accurate δ_C_ predictions by Wiitala and coworkers [52]. More to this point, Stoychev, Auer, and Neese found double-hybrid density functionals, such as DSD-PBEP86, to provide superior performance for δ_H_ predictions that were most comparable to extremely costly coupled cluster calculations [53], while de Oliveria et al. recently reported no benefit for the use of double hybrids over conventional density functionals (i.e., GGAs, meta-GGAs, and hybrids) [54]. Additionally, most authors found that the gauge-independent atomic orbital (GIAO) method provides the best accuracy, but there have been two contemporary reports on improved accuracy using the continuous set of gauge transformations (CSGT) method [49,55]. Thus, it is difficult to choose which model chemistry to apply for spectral data predictions. One possible reason for these discrepancies is the use of vastly different experimental data sets for benchmarking.

One of the earliest and most comprehensive compound sets comprising 80 structures of proton (^1^H) chemical shifts was curated by Rablen et al. [1] in 1999 and later augmented by Tantillo and coworkers [3] with the inclusion of carbon (^13^C) chemical shift data (Figure 1a) and an additional 24 compounds in a separate probe set (Figure 1b). Although the test and probe sets of Rablen and Tantillo have been used for several benchmarking studies [1,3,50,56] with various density functionals, basis sets, and solvent effect studies, they are not without challenges: (1) relativistic effects from elements in rows three and beyond of the periodic table must be included for accurate δ_C_ predictions of carbons bound to the heavy atom (Figure 1c) [57]; (2) molecules such as furfural (Figure 1c) and dimethyl acetal exhibit multiple low-lying conformers in solution whose Boltzmann-weighting factors are highly dependent on the level of theory employed; (3) compounds such as methanol and indole contain hydrogen bond donors that will exhibit concentration-, pH-, and temperature-dependent chemical shifts in solution; (4) aromatic, olefinic, and alkynyl compounds may also exhibit concentration-dependent chemical shifts due to aggregation from π–π stacking [58,59]; and 5) experimental data were not measured in a single solvent system (instead, either CDCl_3_ or CCl_4_ was used, which may yield significant differences [60]). Experimental NMR chemical shift data used by Rablen and Tantillo were from eight sources [61,62,63,64,65,66,67,68] with sample concentrations for carbon NMR spectra, which at times reached up to 10% weight/volume (solid) or volume/volume (liquid). At such high concentrations, interactions between solute molecules may also lead to chemical shift differences [58].

**Table 1 molecules-28-02449-t001:** Comparison of different benchmark studies for ^1^H and ^13^C NMR chemical shift predictions.

Best δ_H_ Method ^a^	Best δ_C_ Method ^a^	Orig. ^b^	Geom. Optimization	Solv./Model ^c^	Conv. ^d^	Benchmark Set	Ref.
mPW1LYP/6-311+G(2d,p)	WP04/DGTZVP	GIAO	B3LYP/6-311+G(2d,p)	CDCl_3_/SMD	linear	104 small organics	[50]
BMK/6-311G(d)	BMK/6-31G(d)	GIAO	B3LYP/6-31+G(d,p)	toluene/none	linear	37 small organics	[48]
B97-2/pcS-3	B97-2/pcS-3	GIAO	B3LYP-D3/def2-TZVP	water/CPCM	MOSS	176 metabolites	[69]
B3LYP/6-31G(d,p)	B3LYP/6-31G(d,p)	GIAO	B3LYP/6-31G(d,p)	gas/none	linear	28 small organics	[70]
B97-2/pcS-2	B97-2/6-311G(d,p)	GIAO	CCSD(T)/cc-pVTZ	gas/none	TMS	29 CCSD(T) calcs.	[71]
δ_H_ not evaluated	B3LYP/6-311+G(d) B3LYP/MIDI!	GIAO	B3LYP/MIDI!	gas/none CDCl_3_/none	linear	15 gas cmpds. 37 solution cmpds.	[72]
not recommended ^e^	LC-TPSS/cc-pVTZ	CSGT	LC-TPSS/def2-SVP	CDCl_3_/COSMO	linear	39 small molecules	[55]
WP04/pcS-2 PBE0/6-31G(d)	PBE0/pcS-2 PBE0/6-31G(d)	GIAO	δ_H_: B3LYP/6-31(d) δ_C_: ωB97X-D/6-31G(d)	CDCl_3_/PCM	linear	24 heterocycles	[51]
WP04/aug-cc-pVDZ	mPW1PW91/ 6-311+G(2d,p)	GIAO	B3LYP/6-31+G(d,p)	CDCl_3_/PCM	linear	23 small organics	[73]
B3LYP/6-311++G(2df,p)	δ_C_ not evaluated	GIAO	B3LYP/6-31+G(d)	CDCl_3_/none	linear	80 small organics	[1]
WP04/aug-cc-pVDZ	δ_C_ not evaluated	GIAO	B3LYP/6-31G(d)	CDCl_3_/PCM	linear	80 small organics	[56]
B3LYP/cc-pVDZ	B3LYP/cc-pVDZ	GIAO	B3LYP/6-31G(d)	CDCl_3_/COSMO	linear	312 small molecules	[74]
SSB-D/ET-pVQZ	SSB-D/ET-pVQZ	GIAO	SSB-D/ET-pVQZ	gas/none	TMS	33 small molecules	[75]
PBE0/cc-pVTZ	PBE0/aug-cc-pVDZ	CSGT	B3LYP/6-311++G(d,p)	CDCl_3_/none	TMS	25 small organics	[49]
B3LYP/6-311++G(d,p)	δ_C_ not evaluated	GIAO	B3LYP/6-31G(d,p)	CDCl_3_/none	C_6_H_6_	14 aromatics	[76]
δ_H_ not evaluated	B3LYP/cc-pVDZ	GIAO	B3LYP/6-311++G(2d,p)	DMSO/CPCM	linear	51 organics	[77]
LH20t/pcSseg-4	mPSTS/pcSseg-4	curr. ^f^	CCSD(T)/cc-pVTZ	gas/none	TMS	23 small organics ^f^	[78]
DSD-PBEP86/ps4	DSD-PBEP86/ps4	GIAO	CCSD(T)/cc-pVTZ	gas/none	CH_4_	15 gas cmpds.	[53]
mPW1PW91/6-311G(d)	same as δ_H_ method	GIAO	B3LYP/6-31G(d,p)	CDCl_3_/PCM	TMS	25 organics	[79]
revTPSS/cc-pVTZ	δ_C_ not evaluated	GIAO	M06-2X/6-311+G(2d,p)	gas/none	TMS	72 small organics	[54]
DSD-PBEP86/pcSseg-3	MP2/pcSseg-3	GIAO	CCSD(T)/cc-pVQZ	gas/none	N/A ^g^	117 gas cmpds.	[80]

^a^ In cases where two methods are listed, no definitive conclusion on the best model chemistry was provided in the study. ^b^ Method for treating gauge origin dependence. ^c^ Solvent refers to the solvent (or gas phase) that the experimental NMR data were measured in, while model refers to the implicit solvent model used, if any. ^d^ Method to convert isotropic shielding tensors to chemical shifts. Linear = linear scaling factors; MOSS = motif-specific scaling (six linear scaling factors for different functionalities); TMS, C_6_H_6_, and CH_4_ = uses these compounds as single point references. ^e^ Errors with respect to experiment were found to be too large for δ_H_ predictions. ^f^ Benchmarking relative to theoretical results from CCSD(T)/def2-TZVP; gauge–origin invariance from current density approach [81]. ^g^ Benchmarking relative to absolute shieldings obtained from high-level in vacuo GIAO-CCSD(T)/pcSseg-3//CCSD(T)/cc-pVQZ calculations.

**Figure 1 molecules-28-02449-f001:**
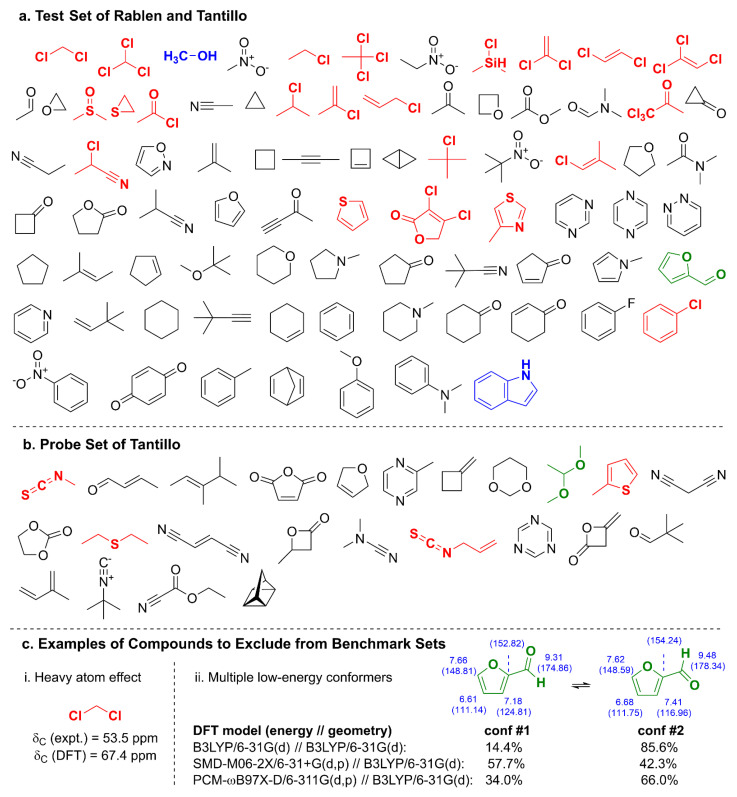
(**a**) Benchmark test set used by Rablen et al. [1] and Tantillo et al. [3] to generate linear scaling factors per Equation (1). Bold-faced, colored structures are modeling challenges due to row three atoms (red), H-bond donors (blue), or multiple low-energy conformers (green). (**b**) Probe set used by Tantillo et al. [3] to examine accuracy of DFT model chemistries for δ_H_ and δ_C_ predictions. (**c**) Examples of problematic structures for benchmarking: (i) attachment of Cl, a row three heavy atom, to carbon results in a 13.9 ppm δ_C_ overprediction by DFT; and (ii) difficulties with accurately determining conformer energetics, as shown for several different energy calculations both with and without dispersion corrections using the same B3LYP/6-31G(d) geometries, can lead to vastly different Boltzmann populations and chemical shift predictions. Both ^1^H and ^13^C chemical shifts (δ_C_ in parentheses) were calculated using the DP4+ model (PCM-mPW1PW91/6-31+G**//B3LYP/6-31G*) [82].

When considering curation of a well-behaved test set, several compounds should be eliminated from the test set for the reasons indicated above (i.e., colored structures in Figure 1), such as those that contain row three elements or hydrogen bond donors or exhibit multiple low-lying conformers. Moreover, the experimental NMR data should be fully verified, including peak assignments, at a sufficiently low concentration in a single solvent system (*viz*., CDCl_3_), and spectra should be appropriately and consistently referenced to ensure reliability. In addition, because several molecules in the test set are highly reactive and thus not easily obtainable (i.e., bicyclobutane, cyclopropanone, and cyclobutene), they should also be excluded from consideration.

An additional plausible reason for the differing conclusions from various DFT chemical shift benchmarking studies is the varying referencing schemes used to convert isotropic shielding tensors into chemical shifts. The simplest method of subtracting the calculated isotropic shielding tensor, σ, for tetramethylsilane (TMS), according to the equation, δ = σ_TMS_ − σ_cmpd_, is also the least accurate due to poor cancellation of errors, especially when comparing carbon atoms with different hybridization [83]. A better approach that improves error cancellation is to replace TMS with a structurally similar reference compound relative to the molecule of interest [84,85,86,87]. Alternatively, the most frequently applied methodology, which also improves error cancellation over TMS and does not require experimental data for specific reference compounds, is to calculate and apply linear scaling factors for a given computational model according to Equation (1) [1,3]. However, the downside is that such linear scaling factors must be determined from a large training set of experimental data, often as part of a comprehensive benchmarking study.
(1)$$\delta =\frac{\text{y-intercept}-\sigma }{-\mathrm{slope}}$$

The goal of this research was to re-measure and verify experimental NMR assignments for a set of compounds, hereafter referred to as DELTA50, that can be used to provide a reliable benchmark for existing and new DFT methodologies as they are developed. In addition, a majority of currently available functionals, basis sets, and solvation models as implemented in Gaussian 16 [88] were tested to determine which is the most appropriate for ^1^H and ^13^C NMR chemical shift predictions. The DELTA50 set was then used to calculate (per Equation (1)) linear scaling factors for conversion of isotropic shielding tensors to chemical shifts. Finally, the performance of the optimal methods and linear scaling factors were assessed using 20 natural products and small- to medium-sized organic compounds, which represents a probe set.

## 2. Results and Discussion

### 2.1. DELTA50 Compound Curation and Experimental Measurements

DELTA50 (Figure 2) contains 50 compounds, comprising 114 proton and 143 carbon chemical shifts, which is a subset of the compounds from the test and probe sets of Rablen [1] and Tantillo [3] (Figure 1) that avoids the previously mentioned problem cases. A wide variety of functional groups (nitro, fluoro, nitrile, aldehyde, ketone, ester, alkyne, amide, amine, olefin, aliphatic, aromatic, and ether), ring sizes (three- to six-membered), and bond hybridizations are present, resulting in ^1^H and ^13^C NMR chemical shifts ranging from 0.25 to 9.80 ppm and −2.9 to 219.4 ppm, respectively. 

Proton and carbon NMR spectra were acquired at 298 K for ≤10 mM solutions of each compound dissolved in CDCl_3_ with 0.03% TMS as internal reference using a 600 MHz spectrometer. A concentration of 10 mM in a 5 mm NMR tube (~1–2 mg) allowed for collection of ^13^C NMR spectra in reasonable overall acquisition times (e.g., approximately 2–4 h on a Bruker 600 MHz AVANCE III spectrometer equipped with a liquid N_2_-cooled broadband Prodigy™ probe) with all resonances detectable (signal-to-noise ratio ≥ 3:1). Proton NMR spectra were acquired using a 2.73 s acquisition time, 20 ppm spectral width, 6.175 ppm transmitter frequency, 16 scans, and a 1 s relaxation delay. Carbon NMR spectra were acquired using a 0.9 s acquisition time, 240 ppm spectral width, 100 ppm transmitter frequency, 3072 to 4096 scans, and a 2 s relaxation delay. Spectral data were processed in MestReNova, version 14.2. using 0.3 and 1.0 Hz exponential line broadenings applied to ^1^H and ^13^C, respectively. Both proton (δ_H_) and carbon (δ_C_) chemical shifts were referenced to TMS at 0.00 ppm and recorded to two decimal places. Individual proton and carbon spectra for each molecule are provided in the Appendix A with expansions for individual multiplets or congested spectral regions, and the assigned chemical shifts are shown for each spectrum; these are also shown in Figure 2.

A concentration-dependent study was undertaken to ensure that samples were sufficiently dilute to avoid deleterious aggregation using a representative subset of compounds with diverse functionalities from DELTA50: benzene, pyridazine, tetrahydrofuran (THF), 3-butyn-2-one, and fluorobenzene (Figure 3). Here, it was found that proton chemical shifts were stable within a variance of ±0.3% at concentrations up to 50 mM. The largest concentration effects were observed for pyridazine and 3-butyn-2-one, which were also found to exhibit the largest proton chemical shift deviations from theoretical predictions in Rablen’s study [1]. THF did not show any chemical shift changes up to a concentration of 50 mM, as would be expected for an aliphatic compound, while the chemical shifts of benzene and fluorobenzene deviated by less than 0.02%, indicating that π–π interactions were not appreciable at these concentrations. The alkyne proton chemical shift (3.21 ppm) of 3-butyn-2-one proportionally increased up to 0.1% with increasing concentration, while the methyl proton resonance (2.38 ppm) showed no change, indicating slight π interactions. The protons meta to nitrogen of pyridazine (7.49 ppm) showed the largest concentration dependence, and it was also noticed in the pyridazine samples that the peak width and chemical shift of residual water varied considerably (υ_1/2_ 12 to 23 Hz and δ_H_ 1.54 to 1.80 ppm). This pointed to the presence of DCl in CDCl_3_ that could protonate basic nitrogens, especially at low concentrations. Thus, a few solid crystals of anhydrous K_2_CO_3_ were added to the CDCl_3_ solvent to neutralize DCl, and this resulted in more consistent chemical shifts as well as markedly improved pyridazine peak shape. Based on these results, anhydrous K_2_CO_3_ was used for all sample preparations of basic compounds, and a 10 mM concentration was considered acceptable for the compounds in the DELTA50 test set.

In some cases where chemical shift assignments were not sufficiently obvious from chemical shifts, scalar coupling, and integration, additional one- and two-dimensional (NOE, gCOSY, multiplicity-edited gHSQC, and/or gHMBC) spectra were acquired for confirmation. In particular, selective NOEs (300 to 700 ms mixing times) were used to determine the assignment of the methyl peaks in 2-methyl-2-butene, *N*,*N*-dimethylacetamide (DMAc), and *N*,*N*-dimethylformamide (DMF).

The differences between the previous experimental data and those recorded in the present study are shown as histogram plots (Figure 4). While 76 proton chemical shifts (67%) were within ±0.05 ppm of the previously reported experimental data, there were 19 outliers (17%) with greater than a ±0.10 ppm deviation. The largest difference of 0.21 ppm was for the α-ether protons of tetrahydropyran. Linear scaling factors calculated at the B3LYP/cc-pVTZ//B3LYP/6-31G(d) level using the previous experimental data were approximately 0.2% different than using the newly acquired data (slope = −1.0429, intercept = 31.6499 versus slope = −1.0450, intercept = 31.6889, respectively), resulting in a maximum 0.04 ppm difference in predicted proton chemical shifts. For δ_C_, 88 out of 143 measurements (62%) were within ±0.25 ppm of the previously reported experimental data; 8 outliers (6%) were found with greater than a ±0.50 ppm difference. The largest carbon chemical shift difference of 2.45 ppm was for the keto carbon of cyclohexanone. Linear scaling factors, at the B3LYP/cc-pVTZ//B3LYP/6-31G(d), differed by approximately 0.1% (slope = −1.0372, intercept = 181.5955 using the previous experimental data, and slope = −1.0364, intercept = 181.5727 using the newly acquired data). This resulted in differences of up to 0.15 ppm.

To ensure that only a single low-energy conformer comprised at least 98% of the Boltzmann distribution, a conformer search was performed for each molecule in the DELTA50 set using a mixed torsional, low-mode sampling search in MacroModel and the OPLS4 force field [89] as implemented in the Schrödinger software suite, version 2021-1 [90]. In some cases, multiple conformers were found, such as boat and chair forms of cyclohexane, and these were analyzed further by DFT energy calculations using the M06-2X/6-31+G(d,p) model chemistry (including vibrational energy corrections) and the SMD solvent model for chloroform, as implemented in Gaussian 16 [88]. This model chemistry was chosen because the M06-2X functional combined with the SMD solvent model has been shown to be particularly effective for prediction of relative energies [91,92,93,94]. Minimum energy conformations were initially verified by lack of imaginary vibrational modes (i.e., the second derivative matrix of energy with respect to displacement was positive definite). Boltzmann probabilities were calculated (see Appendix A for additional details), and the dominant conformer (≥98% probability weighting) was included in the DELTA50 set (as was the case for the chair conformation of cyclohexane).

For each molecule in the DELTA50 set, geometries were optimized at the B3LYP/6-31G(d) level. The Cartesian coordinates and atom numbering for optimized geometries are provided in the Appendix A along with the experimentally measured chemical shifts. A relatively low level of theory was chosen for molecular geometry optimizations because: (1) geometries are reasonably well predicted at this level; (2) geometry optimization is one of the most time-consuming steps in a DFT calculation; and (3) predicted chemical shift dependencies on molecular geometry should be correctable via linear scaling factors [3]. Moreover, the impact of geometry optimization on overall accuracy was further assessed after evaluating the performance of various functionals, basis sets, and solvent models.

### 2.2. DFT Benchmark Study

The performance of 73 density functionals implemented in Gaussian 16 [88] was evaluated for GIAO isotropic shielding predictions using a large, correlation-consistent, triple-zeta basis set, cc-pVTZ, ultrafine integration grid, and the polarizable continuum model (PCM) for solvent effects. Hartree–Fock (HF) theory was also evaluated for comparison purposes. A wide variety of different classes of functionals (Table 2) were included: those based on either the local density approximation (LDAs), generalized gradient approximation (GGAs), meta-GGAs, or hybrids, such as the popular B3LYP functional, that include an empirically derived contribution of HF exchange. In addition, range-separated functionals, which are often denoted with the prefix ‘LC’ for long-range correction, were also evaluated. Range-separated functionals vary the contributions from the various exchange terms based on pairwise electronic spatial distance to overcome problems from long-range density over-delocalization [95]. Finally, the impact of including an empirical dispersion correction, denoted as either ‘D’ or ‘D3′, was also considered. While most of these functionals were designed for reproduction of electronic energies, two functionals, WP04 and WC04, were specifically parameterized for accurate proton and carbon chemical shift prediction, respectively [52].

Isotropic shielding tensors, σ, were converted to chemical shifts through linear regression analysis in Excel. To evaluate performance of each functional, residuals (deviations) were plotted as a function of chemical shift, and root-mean-square deviations (RMSD) and maximum deviations (MD) were calculated (Table 2). Particular focus was placed on evaluation of systematic errors with respect to hybridization and functional groups.

In general, a systematic improvement in accuracy was seen in going from HF theory to LDAs and then to more advanced functionals, e.g., HF < LDAs < GGAs ≈ meta-GGAs < hybrids. However, range separation showed minimal improvement in accuracy, and in most cases, except ωB97X-D, no difference was observed upon including an empirical dispersion correction. Maximum deviations for proton and carbon chemical shifts were typically from 0.2 to 0.5 ppm and 5 to 8 ppm, respectively. HF, SOGGA11, and several of the Minnesota functionals performed particularly poorly. The best performing functionals in the present study were found to be ωB97X-D for carbon chemical shifts and WP04 for proton chemical shifts. 

Figure 5 shows an example of carbon chemical shift deviations from experiment for several representative functionals from each class versus Hartree–Fock theory. Deviations were typically smaller for proton and carbon chemical shifts that resonate upfield (i.e., less than approximately 4.5 and 60 ppm for δ_H_ and δ_C_, respectively), which corresponds to sp^3^-hybridized carbons. The largest deviations for carbon chemical shift predictions were typically carbonyls, olefins, and sp-hybridized carbons. Systematic errors were observed among several functional groups across the board for most density functionals. For instance, mPW1PW91, PBE0, B3LYP, and B97-D overpredicted the more electron-rich carbon in olefins by 2–5 ppm and underpredicted the amide carbonyl in DMF and DMAc by 3–6 ppm. Alkynes and nitriles were overpredicted by several ppm for mPW1PW91 and PBE0, underpredicted by 1–2 ppm for B97-D, or exhibited no noticeable bias for B3LYP.

Following functional evaluation, the basis set size was investigated. In principle, larger, more complete basis sets should have the flexibility to better approximate the electronic density and thereby yield higher accuracy, but this has not always been found to be the case with DFT-based methods [96,97,98]. Importantly, the size of the basis set can dramatically increase computational time. A medium-sized basis set would be most appropriate for larger molecules of greater than 600–700 Da, while the most accurate basis set should be utilized for calculations of small molecules or fragment structures and for critical compounds, such as newly discovered natural products with unusual scaffolds (e.g., homodimericin A [99]). Table 3 shows the errors and computation times for 40 basis sets from double to quadruple zeta, including polarization and diffuse functions on both heavy atoms and hydrogens. While counterintuitive (yet similar to other benchmark studies), smaller basis sets were found to provide more accurate carbon chemical shift predictions [48,51,71], with def2-SVP identified as the best performing basis set when paired with the ωB97X-D functional. Conversely, proton chemical shift predictions typically require larger basis sets, and diffuse functions appear to be productive for reducing errors. The best performing basis set was 6-311++G(2df,p); however, the computational time was found to be more than six times longer for calculation of nitromethane compared to the next smaller Pople-type basis set, 6-311++G(2d,p). Because 6-311++G(2d,p) performed nearly as well, this was chosen as the preferred basis set when paired with WP04 for proton chemical shift predictions.

In general, δ_H_ predictions exhibited substantially fewer systematic errors than δ_C_. There were still a few notable cases: the aldehyde proton of DMF was often underpredicted by 0.1 to 0.5 ppm, cyclopropane was overpredicted by 0.1 to 0.3 ppm, and the alkyne protons in *t*-butyl acetylene and 3-butyn-2-one were underpredicted by 0.1 to 0.3 ppm.

Chemical shift calculations require that the Hamiltonian include interaction terms from the external magnetic field vector, which under a finite basis set leads to a dependence on the choice of the vector origin or gauge [100]. While all gauge methods converge to the same limit at increasing basis set size, most chemical shift calculations are typically handled via GIAO [101,102,103,104,105] due to faster convergence; however, Iron [55] found that the CSGT [105,106,107] method provides more accurate results when paired with long-range corrected (LC) functionals (i.e., the CSGT method with LC-TPSS/cc-pVTZ and the COSMO solvation model was recommended). To test if alternative gauge procedures showed improvement in predictability, CSGT and individual gauges for atoms in molecules (IGAIM) [106,107] were compared to GIAO. Because the accuracy of the gauge method is dependent on basis set size, the basis set was also varied from double- to triple-zeta with inclusion of varying amounts of diffuse and polarization functions. Table 4 shows that, as expected, GIAO converged much more quickly than CSGT and IGAIM. For carbon chemical shift predictions, GIAO produced the lowest RMSD error at only a double-zeta basis set, def2-SVP, and predictions became worse at larger basis sets. The CSGT and IGAIM predictions were nearly equivalent and required triple-zeta basis sets augmented with diffuse functions (viz., aug-cc-pVTZ and jul-cc-pVTZ), which were more than an order of magnitude longer in computation time, to yield comparable levels of accuracy to GIAO with def2-SVP. For proton chemical shifts, GIAO was slower to converge compared to carbon, yet GIAO still exhibited significantly improved accuracy at smaller basis set sizes versus CSGT and IGAIM. Based on these results, GIAO should be the preferred method when considering both speed and accuracy.

Chemical shifts strongly depend on the molecular geometry and internuclear bond distances. Fortunately, the ground state geometry of most compounds has been shown to be well predicted at relatively low levels of theory using a wide variety of density functionals and even Hartree–Fock theory. At the start of this benchmarking study, geometries were optimized in vacuo at the B3LYP/6-31G(d) level, which is a frequently used methodology for computations of spectroscopic properties as well as energetics. The appropriateness of that choice was investigated by holding the NMR chemical shift calculation method constant [PCM-ωB97X-D/def2-SVP for δ_C_ and PCM-WP04/6-311++G(2d,p) for δ_H_] while varying the geometry optimization method. Four different functionals (B3LYP, B3LYP-D3, M06-2X, and ωB97X-D), which are often used for geometry optimizations, were tested when paired with Pople-type basis sets from double- to triple-zeta. In addition, implicit solvation models, PCM or SMD, were applied to the optimizations of B3LYP, B3LYP-D3, and M06-2X because they exhibited the best accuracy for gas phase calculations. Finally, several computationally economical methods, such as HF, PBE, BLYP, and two semi-empirical methods, PM7 and AM1, were also investigated. Data in Table 5 show that the choice of geometry optimization method led to similar accuracy, which validates the choice of using a relatively low level of theory for the bulk of this benchmarking study. B3LYP performed better than M06-2X and ωB97X-D. Inclusion of dispersion correction resulted in a slight improvement in accuracy, at the expense of a negligible increase in computational time. Thus, B3LYP-D3 was used rather than B3LYP, with the additional benefit that the impact of the D3 correction should also enhance the accuracy of the energy prediction for Boltzmann-weighting. Adding an implicit solvation model also resulted in a moderate improvement in accuracy, with the PCM model slightly outperforming SMD. The most accurate predictions for carbon chemical shifts were found when using PCM-B3LYP-D3 with the 6-311G(d,p) basis set, while proton chemical shifts were very slightly less accurate with that basis set compared to 6-31G(d,p).

Finally, the impact of the solvent model on the accuracy of chemical shifts was studied. Three implicit solvation models implemented in Gaussian 16 were tested with the DELTA50 set using either ωB97X-D/def2-SVP for δ_C_ or WP04/6-311++G(2d,p) for δ_H_. The integral equation formalism (IEF) version of the polarizable continuum model (PCM) is the recommended (default) model in Gaussian 16 [108]. The SMD model of Truhlar et al. [109] is a revised version of the PCM model that was developed specifically for reproducing solvation energies. Finally, the polarizable conductor calculation model, CPCM, [110,111] was also evaluated. Results in Table 6 show a marked improvement with the use of any solvation model but only a slight improvement in accuracy when using PCM versus CPCM or SMD.

**Table 6 molecules-28-02449-t006:** Impact of implicit solvent model on chemical shift predictions ^a^.

	δ_H_ (ppm)		δ_C_ (ppm)	
Solvent Model ^b^	RMSD ^c^	MD ^c^	RMSD ^c^	MD ^c^
PCM	0.079	0.21	1.50	4.61
CPCM	0.080	0.20	1.50	4.57
SMD	0.087	0.29	1.51	4.69
none	0.107	0.33	1.84	5.31

^a^ Calculations using PCM (chloroform) solvent model and B3LYP/6-31G(d) geometries. ^b^ Recommended implicit solvation model highlighted in blue, bold font. ^c^ RMSD = root-mean-square deviation; MD = maximum deviation.

Based on the totality of the previous results, the best performing density functional for carbon chemical shift prediction was found to be ωB97X-D when paired with the def2-SVP basis set. For proton chemical shift predictions, the WP04 functional exhibited the lowest error when combined with the 6-311++G(2df,p) basis set, but calculation times were unreasonably long. In contrast, the smaller 6-311++G(2d,p) basis set gave nearly comparable accuracy at a six-fold reduced computational cost. For both proton and carbon chemical shift predictions, the GIAO method was most accurate at these basis set sizes. Molecular geometries should be optimized at the B3LYP-D3/6-311G(d,p) level. Implicit solvent effects from the PCM model should be included at all stages of the calculation. The best performing models were δ_H_: GIAO-PCM-WP04/6-311++G(2d,p)//PCM-B3LYP-D3/6-311G(d,p) and δ_C_: GIAO-PCM-ωB97X-D/def2-SVP//PCM-B3LYP-D3/6-311G(d,p). When it is necessary to reduce calculation times for larger molecules, the geometry optimization step can be changed to B3LYP/6-31G(d) with a moderate reduction in accuracy for carbon chemical shift predictions while maintaining the same level of accuracy for protons after changing the basis set to jul-cc-pVDZ for proton NMR calculations. Importantly, dispersion corrections should still be used for electronic energy calculations with this faster method. In cases where dispersive interactions may be critical to the optimized geometry, such as inclusion of explicit solvent and studies of organometallic complexes, then the high-accuracy method should be used. The recommended methods and linear scaling factors are listed in Table 7.

**Table 7 molecules-28-02449-t007:** Recommended DFT methods and linear scaling factors.

Calculation Step	Method 1: Speed + Efficiency	Method 2: High Accuracy
Geometry Optimization	B3LYP/6-31G(d) ^a^	PCM-B3LYP-D3/6-311G(d,p)
Energy Calculation	PCM-B3LYP-D3/6-31G(d)	PCM-B3LYP-D3/6-311G(d,p)
δ_H_ Calculation	GIAO-PCM-WP04/jul-cc-pVDZ	GIAO-PCM-WP04/6-311++G(2d,p)
δ_H_ Scaling Factors ^b^	m = −1.0309, b = 31.8883	m = −1.0311, b = 32.2654
δ_C_ Calculation	GIAO-PCM-ωB97X-D/def2-SVP	GIAO-PCM-ωB97X-D/def2-SVP
δ_C_ Scaling Factors ^b^	m = −1.0081, b = 195.6683	m = −1.0065, b = 196.0386

^a^ The “fine” integration grid will also improve calculation speed with a negligible impact on accuracy at this level of theory (note: a larger grid is needed for meta-GGAs [112]). ^b^ slope = m, y-intercept = b; to be used in Equation (1).

### 2.3. Probe Set Evaluation

Next, the performance of each method from Table 7 was evaluated for chemical shift predictions of small- to medium-sized complex synthetic organic compounds and natural products. In this fashion, DELTA50 can be considered a training set for generation of linear scaling factors or optimization of new empirical density functionals, while the series of complex structures and natural products represents a probe set. Twenty relatively rigid, well-studied compounds ranging in molecular weights from 96 to 854 g mol^−1^ with experimental NMR data measured in deuterated chloroform were included in this probe set. These structures and calculation results are listed in Figure 6 and Table 8, respectively. Most of the compounds in the probe set have existing single-crystal X-ray diffraction (SCXRD) data, and/or their structures have been verified via total synthesis. Additionally, for 18 out of 20 compounds, previous DFT chemical shift calculations have been performed. For several cases, multiple high-quality computational NMR studies have been performed (see Appendix A for data from additional reports). This allows for direct comparison to the best performing methods in this study. Also noteworthy is that for most of these compounds, extensive 2D NMR data have been collected, such that the proton and carbon chemical shift assignments are unlikely to be misassigned. A few proton and carbon assignments are still ambiguous, such as the carbons resonating from 40 to 50 ppm of ingenane diterpene 8, and these have been left out of the probe set (further details can be found in the Appendix A).

Several trends are immediately noticeable. As expected, method two generally exhibited the most accurate performance as evident by the lowest RMSD for 14 out of 20 proton chemical shift predictions and 16 out of 20 carbon chemical shift predictions. Moreover, the maximum δ_C_ deviation across all compounds, representing 424 carbon chemical shift predictions, was only 6.5 ppm for a ketone carbon of homodimericin A, which provides a relatively low upper limit upon which putative NMR structure proposals could be called into question following DFT calculations using the prescribed model chemistry in this paper. It should also be noted that in a few cases where existing literature DFT methods were found to outperform method two, such as the δ_H_ predictions of strychnine or the δ_C_ predictions of artemisinin, these were from studies where the DFT procedures were specifically optimized for these classes of compounds rather than being a general-purpose chemical shift model chemistry for organic compounds.

From Figure 6, there also appears to be a spatial relationship between the least accurate predicted proton and carbon chemical shift from method two, which is highlighted by the colored circles. This was unexpected considering that markedly different optimal density functionals and basis set sizes were used for carbon (GIAO-PCM-ωB97X-D/def2-SVP) versus proton [GIAO-PCM-WP04/6-311++G(2d,p)] chemical shift predictions, although they share the same geometry optimization method [PCM-B3LYP-D3/6-311G(d,p)]. In all cases except echinopine B, the least accurately predicted proton is not attached to the least accurately predicted carbon. Rather, while both nuclei are in close spatial proximity, they are separated by approximately 2.5 to 5 Å. This is most pronounced for olefinic carbons that are often overpredicted by approximately 3 to 5 ppm leading to an *under*prediction, or *over*-shielding effect, on nearby protons by approximately 0.1 to 0.3 ppm. It is also noteworthy that a few investigators have found that isotropic shielding tensors are particularly sensitive to the molecular geometry [113,114], and this may indicate that future improvements in density functional performance for chemical shift predictions might come indirectly from improvements in the geometry optimization method rather than from the functional used for GIAO shielding tensor calculations.

**Table 8 molecules-28-02449-t008:** DFT method performances for chemical shift predictions of probe compounds.

Compound	MW (g mol^−1^)	Confs ^c^	δ (ppm)	This Study ^a^				Previous DFT Studies		
				Method 1		Method 2				
				RMSD ^d^	MD ^d^	RMSD ^d^	MD ^d^	RMSD ^d^	MD ^d^	Ref. ^e^
bicyclo [2.1.1]-hexan-2-one	96	1	δ_H_: δ_C_:	**0.03** 0.6	**0.05** 0.8	0.07 **0.5**	0.12 **0.7**	0.12 1.1	0.20 1.7	[26]
α-pinene	136	1	δ_H_: δ_C_:	**0.08** 1.6	0.19 **3.5**	** 0.08 ** ** 1.5 **	**0.15** 3.6	0.63 3.6	1.14 7.4	[115]
aquatolide	246	3	δ_H_: δ_C_:	0.10 1.7	0.26 4.1	** 0.05 ** ** 1.5 **	** 0.11 ** ** 3.2 **	0.11 1.8	0.27 4.1	[26]
naupliolide	246	4	δ_H_: δ_C_:	0.14 2.3	0.33 **5.5**	** 0.10 ** ** 1.9 **	**0.20** 5.6	0.23 3.0	0.58 7.8	[26]
echinopine B	246	9	δ_H_: δ_C_:	** 0.07 ** ** 1.4 **	**0.17** 2.6	0.08 **1.4**	0.22 **2.5**	0.10 2.5	0.22 5.5	[26]
parthenolide	248	3	δ_H_: δ_C_:	0.08 1.5	0.14 3.0	** 0.02 ** ** 1.4 **	** 0.04 ** ** 2.8 **	not available		
diepoxy-guaianolide	262	5	δ_H_: δ_C_:	0.12 1.7	0.21 4.1	**0.05** 1.8	**0.12** 4.1	0.18 **1.3**	0.44 **2.6**	[116]
cannabicitran (CBT-C)	258 ^b^	2	δ_H_: δ_C_:	0.11 **0.9**	**0.17** 2.2	**0.07** 1.0	**0.17** 2.2	0.12 1.1	0.31 **2.0**	[117]
ingenane diterpene 8	278	2	δ_H_: δ_C_:	0.17 2.7	0.39 4.5	** 0.06 ** ** 2.2 **	**0.14** 4.7	0.08 **2.2**	0.20 **3.9**	[26]
artemisinin	282	1	δ_H_: δ_C_:	**0.10** 1.1	0.25 2.1	0.12 1.1	**0.24** 2.3	-- ^f^ **0.8**^g^	-- ^f^ **1.4 **^g^	[118]
nobilistine A	317	20	δ_H_: δ_C_:	0.14 1.7	**0.25** 3.6	** 0.12 ** ** 1.5 **	**0.25** 3.5	0.27 1.6	0.65 **3.1**	[119]
intricarene	326	2	δ_H_: δ_C_:	0.10 2.3	0.24 4.0	** 0.09 ** ** 2.1 **	** 0.21 ** ** 3.8 **	0.12 2.2	0.27 4.9	[26]
strychnine	334	3	δ_H_: δ_C_:	0.15 1.5	0.41 4.0	0.10 **1.4**	0.25 **3.8**	**0.08** 1.8	**0.18** 6.7	[120]
holstiine	382	4	δ_H_: δ_C_:	0.16 2.4	0.30 7.7	** 0.10 ** ** 1.9 **	** 0.23 ** ** 5.1 **	0.21 2.9	0.47 11.3	[121]
colchicine	399	81	δ_H_: δ_C_:	** 0.10 ** ** 2.1 **	0.21 **3.8**	0.11 2.2	**0.20** 4.0	0.16 2.3	0.25 5.0	[59]
hexacyclinol	416	23	δ_H_: δ_C_:	0.15 2.4	0.38 7.0	** 0.13 ** ** 2.1 **	** 0.30 ** ** 5.9 **	0.29 4.6	0.62 9.0	[122]
homodimericin A	491	9	δ_H_: δ_C_:	**0.10** 3.4	0.21 7.7	** 0.10 ** ** 2.9 **	** 0.19 ** ** 6.5 **	not available		
strychnobaillonine	613	12	δ_H_: δ_C_:	0.19 3.0	0.46 10.4	** 0.16 ** ** 2.4 **	** 0.34 ** ** 6.4 **	0.22 2.9	0.62 6.7	[123]
sungucine	635	11	δ_H_: δ_C_:	0.18 1.9	0.51 **4.4**	** 0.14 ** ** 1.8 **	**0.31** 4.5	0.18 **1.8**	0.64 5.4	[124]
paclitaxel	854	>157	δ_H_: δ_C_:	**0.17** 2.8	**0.43** 7.2	0.19 **2.3**	0.52 **6.3**	-- ^f^ 3.7	-- ^f^ 9.1	[125]

^a^ Results from performance of methods one and two from Table 7. ^b^ For cannabicitran, calculations were performed on a truncated structure (MW 258). ^c^ Number of conformers within 5 kcal mol^−1^ of the DFT-calculated global energy minimum. ^d^ RMSD = root-mean-square deviation; MD = maximum deviation, in ppm. (Best performance for each compound is highlighted in blue, bold font). ^e^ Reference to previous DFT prediction results if available. See SI for calculation details. ^f^ The only DFT predictions available were for δ_C_. ^g^ Only 6 out of 15 δ_C_ calculations were reported [118].

## 3. Materials and Methods

The following solvents and standard reagents were purchased from Sigma-Aldrich: chloroform-*d* (99.96 atom %D) with 0.03% (*v*/*v*) TMS, tetrahydrofuran (anhydrous, inhibitor-free, ≥99%), 1-methyl pyrrole (99%), γ-butyrolactone (99+%), nitroethane (99.5%), pyrazine (99+%), bicyclo [2.2.1]hepta-2,5-diene (98%), cyclopentane (analytical standard), pyridazine (98%), fluorobenzene (analytical standard), pyrimidine (≥98.0%), acetaldehyde (ACS reagent grade, ≥99.5%, in a sealed ampule), trimethylene oxide (97%), 2-butyne (98%), ethylene oxide (2000 μg/mL dissolved in dichloromethane, a certified reference material, in a sealed ampule), furan (≥99%), cyclopropane (≥99%, in a gas cylinder), 2-methyl propene (99%, in a gas cylinder), *N*,*N*-dimethylacetamide (99.8%, extra dry), cyclohexane (99.5%, anhydrous), methyl acetate (99.5%, anhydrous), nitromethane (≥99.0%), acetonitrile (99.8%, anhydrous), *N*,*N*-dimethylformamide (99.8%, anhydrous), 2,5-dihydrofuran (97%), toluene (99.8%, anhydrous), isobutyronitrile (99%), acetone (≥99.9%, HPLC grade), *tert*-butyl methyl ether (99.8%, anhydrous), tetrahydropyran (99%, anhydrous), propionitrile (analytical standard), 2-methyl-2-butene (99+%), and *N*-methylpyrrolidine (97%). 3-Butyn-2-one (98+%) was obtained from Lancaster Synthesis. The following reagent standards were purchased from Oakwood Chemicals: cyclobutanone, cyclopent-2-enone, 3,3-dimethyl-1-butene, 2-methyl-2-nitropropane, benzene (ACS grade), nitrobenzene, 3,3-dimethyl-1-butyne, pyridine, anisole, cyclohex-2-enone, cyclohexanone, 1-methyl piperidine, maleic anhydride, p-benzoquinone, pivalonitrile, cyclopentanone, and isoxazole.

Dissolved solutions of gaseous reagents, cyclopropane and isobutylene (2-methyl propene), were prepared by bubbling them through CDCl_3_ and then serially diluting until acceptable NMR spectra were obtained (i.e., chemical shifts were stable upon dilution, indicating no aggregation).

Homodimericin A was provided by Aili Fan (Peking University, Beijing, China). 

A variety of density functionals, as implemented in Gaussian 16 [88], were evaluated as part of the benchmarking study. The exchange and correlation components of LDAs, GGAs, and meta-GGAs as well as standalone combinations are listed in the Appendix A along with primary references. In addition, the long-range correction method of Hirao and co-workers was also applied to the non-hybrid functionals, BP86, BPW91, N12, TPSS, revTPSS, and M06-L, which were noted in Table 1 with the prefix ‘LC’.

## 4. Conclusions

Experimental proton and carbon NMR chemical shift data for 50 structurally diverse compounds dissolved in deuterated chloroform were carefully measured in this study, enabling a comprehensive benchmark of DFT methods. Linear scaling factors to convert isotropic shielding tensors to chemical shifts were generated for two recommended methodologies that balance speed and accuracy. These two best performing methods were then evaluated against 20 probe organic compounds and natural products with high accuracy observed particularly when compared to previously reported DFT predictions for proton and carbon chemical shifts of these well-studied structures.

Of particular importance, this chemical shift test set can be used for performance evaluation of newly developed model chemistries as well as future design and optimization of empirical density functionals. Such work will be a focus for future research conducted in our laboratory.

## Figures and Tables

**Figure 2 molecules-28-02449-f002:**
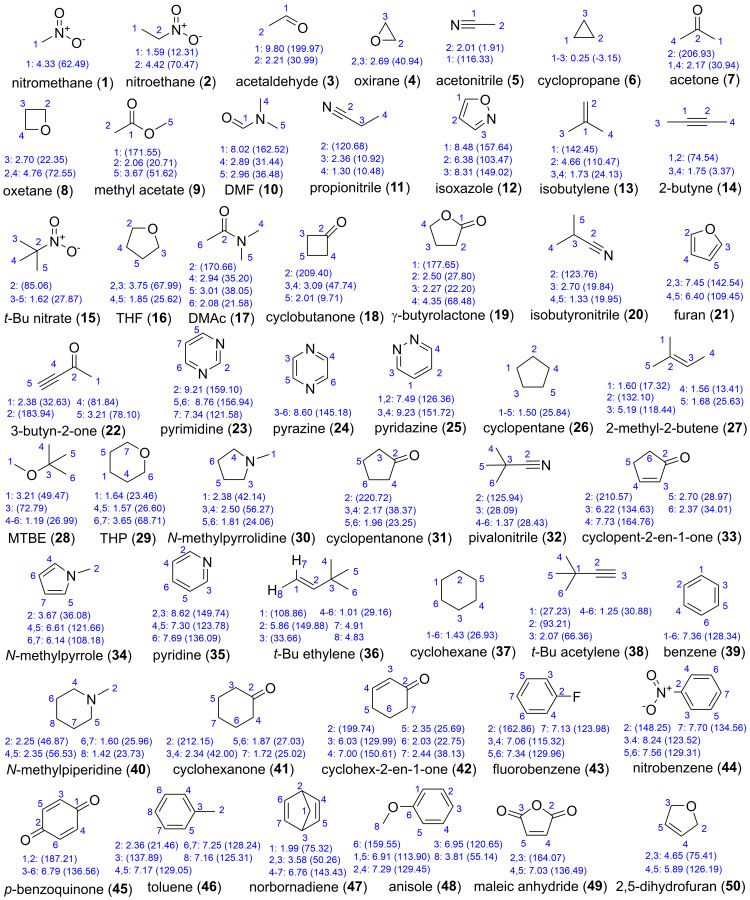
DELTA50 benchmark set of 50 small molecules that avoid the issues described in Figure 1. Experimental ^1^H chemical shifts are provided for numbered carbon atoms with ^13^C chemical shifts listed in parentheses. Numbering corresponds to that used in calculations. Experimental data were acquired on a 600 MHz NMR in CDCl_3_ and referenced to TMS at 0.00 ppm for both ^1^H and ^13^C.

**Figure 3 molecules-28-02449-f003:**
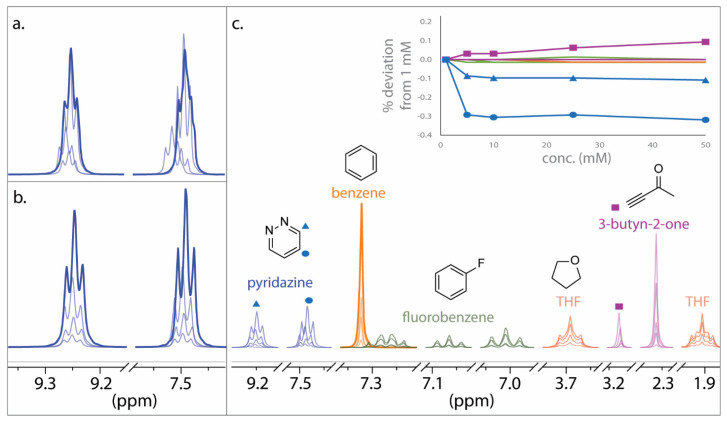
Proton chemical shift dependence as a function of concentration in CDCl_3_. (**a**) Pyridazine without K_2_CO_3_. (**b**) Pyridazine with K_2_CO_3_ to neutralize residual DCl. (**c**) 600 MHz NMR spectral overlays for individual compounds. Concentrations were varied from 1 to 50 mM. The largest chemical shift deviations with respect to concentration are highlighted for pyridazine (blue triangles and circles) and the alkyne proton of 3-butyn-2-one (pink squares).

**Figure 4 molecules-28-02449-f004:**
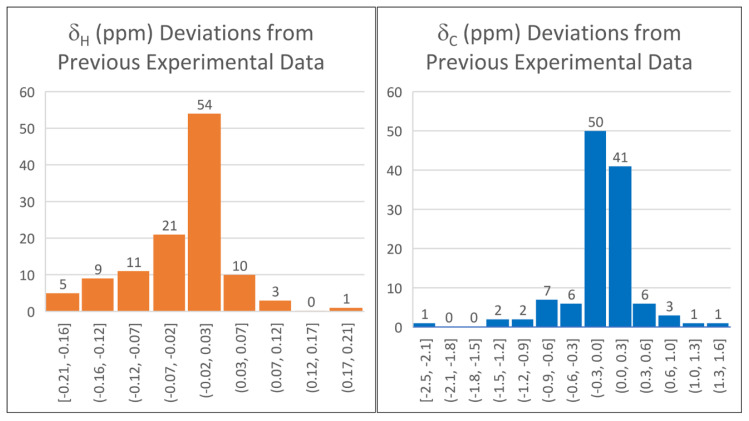
Histogram plots showing frequency of measured differences between experimental data available in the literature compared with re-measured values in this study for 114 proton chemical shifts and 143 carbon chemical shifts.

**Figure 5 molecules-28-02449-f005:**
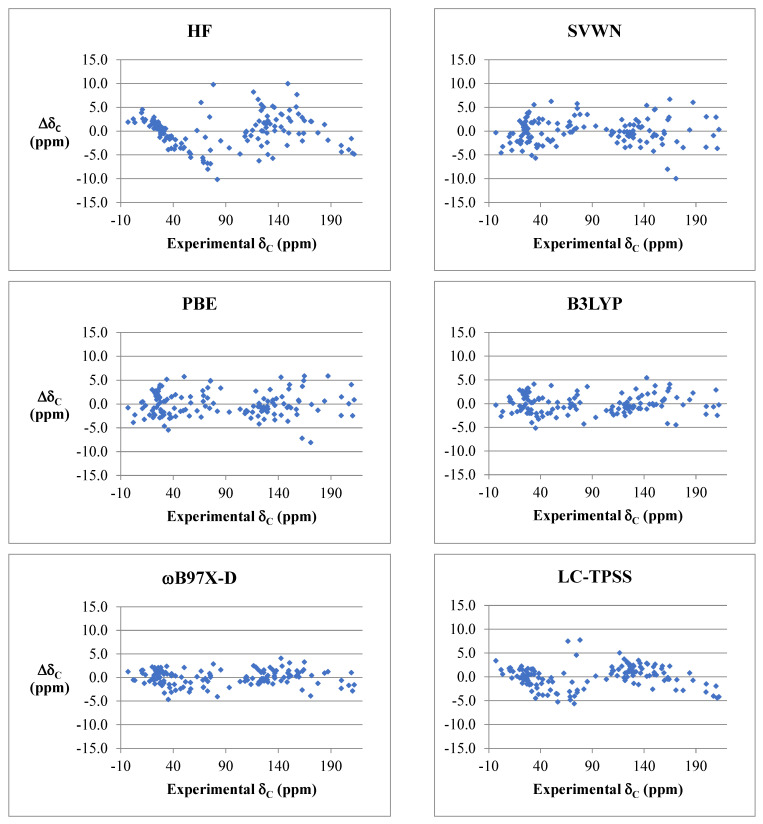
Comparison of deviations for carbon chemical shift predictions for representative density functionals and HF (cc-pVTZ basis sets, B3LYP/6-31G* optimized geometries, and PCM solvation model for chloroform was used in all instances).

**Figure 6 molecules-28-02449-f006:**
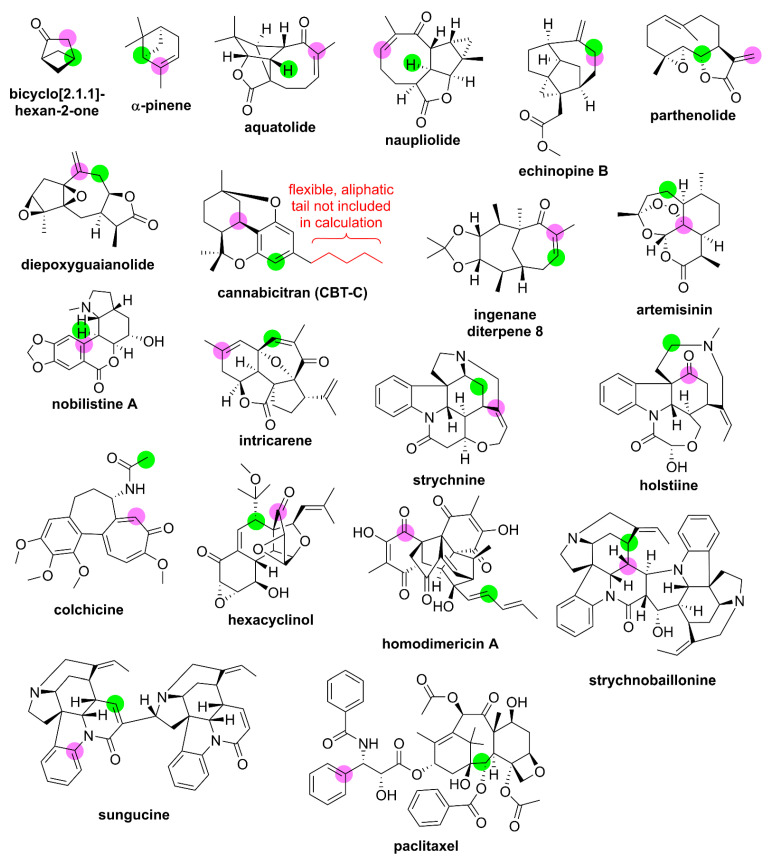
Probe set structures. Atomic locations of maximum chemical shift deviations compared to experiment for method two calculations from this study are highlighted for δ_C_ and δ_H_ in magenta and lime green, respectively.

**Table 2 molecules-28-02449-t002:** Performance of density functionals and HF for ^1^H and ^13^C chemical shift predictions.

	δ_H_ (ppm)		δ_C_ (ppm)			δ_H_ (ppm)		δ_C_ (ppm)	
Functional ^a,b^	RMSD ^c^	MD ^c^	RMSD ^c^	MD ^c^	Functional ^a,b^	RMSD ^c^	MD ^c^	RMSD ^c^	MD ^c^
HF	0.190	0.74	3.44	10.15	TPSS	0.107	0.26	2.38	6.62
Xα	0.152	0.43	2.85	10.03	revTPSS	0.108	0.26	2.32	7.13
SVWN	0.144	0.44	2.68	9.95	PKZB	0.129	0.31	2.65	6.49
BLYP	0.127	0.42	2.86	7.12	BRxBRc	0.130	0.36	2.78	6.61
BP86	0.126	0.37	2.51	7.64	VSXC	0.124	0.35	3.44	10.52
BVP86	0.125	0.37	2.51	7.61	τ-HCTH	0.114	0.37	2.43	5.99
BPW91	0.124	0.39	2.43	7.61	M06-L	0.094	0.24	2.24	6.54
mPWPW91	0.126	0.39	2.51	7.71	M11-L	0.134	0.43	3.63	10.06
PBE	0.132	0.40	2.55	8.06	MN12-L	0.116	0.38	3.04	8.46
SOGGA11	0.169	0.54	4.10	9.43	MN15-L	0.118	0.30	3.10	9.19
SOGGA11X	0.111	0.31	1.67	5.16	LC-TPSS	0.147	0.52	2.33	7.75
BPL	0.115	0.42	2.81	7.25	LC-revTPSS	0.145	0.50	2.26	7.55
G96LYP	0.118	0.40	2.63	6.91	LC-M06-L	0.142	0.55	2.16	6.89
B97-D	0.117	0.36	2.66	6.19	CAM-B3LYP	0.102	0.25	1.66	5.02
B97-D3	0.117	0.36	2.66	6.19	LC-ωPBE	0.139	0.40	1.99	5.65
HCTH	0.127	0.44	2.74	7.05	LC-ωHPBE	0.139	0.40	1.99	5.65
HCTH/93	0.119	0.39	2.57	6.98	ωB97	0.130	0.37	1.78	4.46
HCTH/147	0.121	0.39	2.66	6.84	ωB97X	0.119	0.32	1.62	4.57
N12	0.112	0.39	2.47	6.33	ω B97X-D	0.109	0.29	1.57	4.64
LC-BP86	0.148	0.49	2.40	8.18	HISS	0.126	0.40	1.99	6.38
LC-BPW91	0.152	0.51	2.47	8.34	HSE06	0.109	0.26	1.77	4.89
LC-N12	0.153	0.56	2.52	7.98	N12-SX	0.110	0.28	1.78	4.80
B3LYP	0.098	0.26	1.97	5.49	B1B95	0.113	0.32	1.77	5.22
B3PW91	0.105	0.25	1.77	5.03	TPSSh	0.097	0.22	1.99	5.42
B1LYP	0.096	0.24	1.90	5.45	τ-HCTHhyb	0.101	0.25	1.96	5.24
O3LYP	0.109	0.31	2.20	5.61	M05	0.131	0.34	2.72	10.79
X3LYP	0.098	0.24	1.96	5.54	M05-2X	0.166	0.60	2.72	8.17
mPW1PW91	0.107	0.27	1.72	4.81	M06-2X	0.161	0.57	2.70	7.19
mPW1PBE	0.108	0.27	1.72	4.80	M06-HF	0.295	1.06	6.30	17.26
mPW1LYP	0.097	0.24	1.96	5.61	M08-HX	0.165	0.58	3.28	9.16
mPW3PBE	0.106	0.25	1.80	5.17	MN15	0.142	0.41	2.26	5.93
PBE0	0.109	0.27	1.74	4.85	PW6B95	0.108	0.29	1.80	5.03
PBEh1PBE	0.109	0.27	1.74	4.88	PW6B95-D3	0.108	0.29	1.80	5.03
WP04	0.086	0.32	2.73	10.21	M11	0.180	0.62	3.27	10.21
WC04	0.150	0.42	2.99	8.00	MN12-SX	0.110	0.29	2.44	8.02
B97-1	0.101	0.24	1.85	5.20	APF	0.108	0.26	1.74	4.88
B97-2	0.103	0.23	1.78	4.70	B98	0.099	0.24	1.84	5.21

^a^ Gas phase B3LYP/6-31G(d) geometries, cc-pVTZ basis set, and PCM(CHCl_3_) were used. ^b^ Recommended functionals highlighted in blue, bold font. ^c^ RMSD = root-mean-square deviation; MD = maximum deviation.

**Table 3 molecules-28-02449-t003:** Impact of basis set on accuracy of ^1^H and ^13^C NMR chemical shift predictions ^a^.

Functional:		WP04		ωB97X-D		Functional:		WP04		ωB97X-D	
		δ_H_ (ppm)		δ_C_ (ppm)				δ_H_ (ppm)		δ_C_ (ppm)	
Basis Set ^b^	Time ^c^	RMSD ^d^	MD ^d^	RMSD ^d^	MD ^d^	Basis Set ^b^	Time ^c^	RMSD ^d^	MD ^d^	RMSD ^d^	MD ^d^
SV	0.11	0.171	0.48	2.04	5.90	6-31G(d,p)	0.25	0.098	0.37	1.51	5.91
SVP	0.22	0.119	0.45	1.50	4.61	6-31+G(d,p)	0.31	0.086	0.24	1.59	4.77
TZV	0.18	0.162	0.80	2.53	8.01	6-311G(d,p)	0.31	0.095	0.33	1.80	5.08
TZVP	0.36	0.096	0.32	1.65	4.46	6-311+G(d,p)	0.40	0.086	0.37	1.69	4.54
def2-SV	0.20	0.148	0.47	1.68	5.93	6-311++G(d,p)	0.44	0.086	0.35	1.71	4.60
def2-SVP	0.24	0.119	0.45	1.50	4.61	6-311++G(2d,p)	0.59	0.077	0.30	1.68	4.66
def2-TZV	0.18	0.162	0.80	2.53	8.01	6-311++G(2df,p)	3.76	0.075	0.27	1.54	4.35
def2-TZVP	2.93	0.080	0.25	1.63	4.81	6-311++G(2df,2p)	4.05	0.078	0.29	1.53	4.43
def2-TZVPP	3.63	0.084	0.28	1.61	4.53	apr-cc-pVDZ	0.36	0.096	0.30	1.77	5.18
def2-QZV	0.33	0.147	0.62	2.23	6.11	may-cc-pVDZ	0.36	0.096	0.30	1.77	5.18
EPR-II	0.35	0.122	0.34	2.17	11.34	jun-cc-pVDZ	0.36	0.096	0.30	1.77	5.18
EPR-III	4.52	0.079	0.30	1.60	4.75	jul-cc-pVDZ	0.50	0.079	0.21	2.07	6.99
D95	0.14	0.163	0.49	2.93	13.71	aug-cc-pVDZ	0.70	0.080	0.29	2.12	6.29
D95V	0.13	0.165	0.49	2.97	13.76	apr-cc-pVTZ	4.14	0.082	0.34	1.57	4.66
MIDI!	0.14	0.183	0.68	2.41	6.09	may-cc-pVTZ	4.14	0.082	0.34	1.57	4.66
3-21G	0.09	0.215	0.76	2.19	6.28	jun-cc-pVTZ	5.24	0.082	0.31	1.63	4.89
4-31G	0.11	0.172	0.61	2.28	6.07	jul-cc-pVTZ	8.33	0.081	0.30	1.62	4.85
6-21G	0.11	0.208	0.63	2.19	6.08	aug-cc-pVTZ	12.86	0.081	0.28	1.66	5.12
6-31G	0.11	0.162	0.59	2.08	5.55	cc-pVDZ	0.27	0.109	0.35	1.71	4.96
6-31G(d)	0.20	0.115	0.47	1.62	7.25	cc-pVTZ	3.66	0.086	0.32	1.57	4.64

^a^ Calculations using gas phase B3LYP/6-31G(d) geometries and PCM (chloroform) solvent model. ^b^ Recommended basis sets highlighted in blue, bold font. ^c^ Relative time for PCM-ωB97X-D calculation of nitromethane with specified basis set. ^d^ RMSD = root-mean-square deviation; MD = maximum deviation.

**Table 4 molecules-28-02449-t004:** Impact of gauge-referencing method and basis set on δ_H_ and δ_C_ predictions ^a^.

	Gauge Method ^b^	Functional:	WP04		ωB97X-D	
			δ_H_ (ppm)		δ_C_ (ppm)	
Basis Set		Time ^c^	RMSD ^d^	MD ^d^	RMSD ^d^	MD ^d^
def2-SVP	GIAO	0.24	0.119	0.45	1.50	4.61
def2-TZVP	GIAO	2.93	0.080	0.25	1.63	4.81
def2-TZVPP	GIAO	3.63	0.084	0.28	1.61	4.53
6-31G(d,p)	GIAO	0.25	0.098	0.37	1.51	5.91
6-31+G(d,p)	GIAO	0.31	0.086	0.24	1.59	4.77
6-311+G(d,p)	GIAO	0.40	0.086	0.37	1.69	4.54
6-311++G(d,p)	GIAO	0.44	0.086	0.35	1.71	4.60
6-311++G(2d,p)	GIAO	0.59	0.077	0.30	1.68	4.66
6-311++G(2df,p)	GIAO	3.76	0.075	0.27	1.54	4.35
6-311++G(2df,2p)	GIAO	4.05	0.078	0.29	1.53	4.43
jul-cc-pVDZ	GIAO	0.50	0.079	0.21	2.07	6.99
aug-cc-pVDZ	GIAO	0.70	0.080	0.29	2.12	6.29
jul-cc-pVTZ	GIAO	8.33	0.081	0.30	1.62	4.85
aug-cc-pVTZ	GIAO	12.86	0.081	0.28	1.66	5.12
def2-SVP	CSGT	0.23	0.321	1.62	2.96	10.20
def2-TZVP	CSGT	2.78	0.100	0.41	2.05	6.20
def2-TZVPP	CSGT	3.26	0.088	0.34	1.83	5.84
6-31G(d,p)	CSGT	0.24	0.385	2.29	2.26	7.90
6-31+G(d,p)	CSGT	0.29	0.313	1.89	1.70	5.79
6-311+G(d,p)	CSGT	0.36	0.194	0.92	2.16	6.73
6-311++G(d,p)	CSGT	0.39	0.188	0.88	2.17	6.90
6-311++G(2d,p)	CSGT	0.48	0.087	0.23	1.76	5.03
6-311++G(2df,p)	CSGT	3.47	0.092	0.44	1.78	5.31
6-311++G(2df,2p)	CSGT	3.68	0.082	0.41	1.76	5.35
jul-cc-pVDZ	CSGT	0.41	0.121	0.53	2.06	8.22
aug-cc-pVDZ	CSGT	0.54	0.114	0.51	2.04	8.50
jul-cc-pVTZ	CSGT	6.73	0.080	0.36	1.58	4.54
aug-cc-pVTZ	CSGT	9.63	0.081	0.36	1.56	4.45
def2-SVP	IGAIM	0.23	0.321	1.63	2.96	10.22
def2-TZVP	IGAIM	2.78	0.100	0.41	2.05	6.21
def2-TZVPP	IGAIM	3.18	0.087	0.34	1.83	5.85
6-31G(d,p)	IGAIM	0.24	0.386	2.31	2.26	7.90
6-31+G(d,p)	IGAIM	0.29	0.314	1.91	1.69	5.76
6-311+G(d,p)	IGAIM	0.36	0.194	0.93	2.17	6.73
6-311++G(d,p)	IGAIM	0.39	0.188	0.89	2.18	6.90
6-311++G(2d,p)	IGAIM	0.48	0.087	0.23	1.76	5.03
6-311++G(2df,p)	IGAIM	3.47	0.092	0.44	1.78	5.31
6-311++G(2df,2p)	IGAIM	3.68	0.082	0.41	1.76	5.36
jul-cc-pVDZ	IGAIM	0.39	0.121	0.53	2.06	8.21
aug-cc-pVDZ	IGAIM	0.54	0.114	0.51	2.04	8.50
jul-cc-pVTZ	IGAIM	6.55	0.080	0.36	1.58	4.54
aug-cc-pVTZ	IGAIM	9.66	0.081	0.36	1.56	4.45

^a^ Calculations using gas phase B3LYP/6-31G(d) geometries and PCM (chloroform) solvent model ^b^ Recommended gauge method highlighted in blue, bold font. ^c^ Relative time for PCM-ωB97X-D calculation of nitromethane with specified basis set. ^d^ RMSD = root-mean-square deviation; MD = maximum deviation.

**Table 5 molecules-28-02449-t005:** Impact of molecular geometry on chemical shift predictions ^a^.

NMR Method		PCM-ωB97X-D/def2-SVP		PCM-WP04/6-311++G(2d,p)	
	Time ^c^ (h)	δ_H_ (ppm)		δ_C_ (ppm)	
Geometry Optimization Method ^b^		RMSD ^d^	MD ^d^	RMSD ^d^	MD ^d^
AM1	0.001	0.217	1.24	2.96	9.11
PM7	0.005	0.260	1.61	2.32	8.56
HF/MIDI!	0.105	0.094	0.41	1.65	5.51
HF/6-31G(d)	0.149	0.103	0.38	1.94	5.77
BLYP/6-31G(d)	0.286	0.080	0.29	1.76	7.26
PBE/6-31G(d)	0.295	0.080	0.23	1.61	5.37
B3LYP/3-21G	0.152	0.104	0.48	2.35	6.97
B3LYP/MIDI!	0.208	0.086	0.37	1.83	5.56
B3LYP/6-31G(d)	0.284	0.078	0.30	1.50	4.61
B3LYP/6-31G(d,p)	0.368	0.077	0.30	1.49	4.55
B3LYP/6-311G(d,p)	0.624	0.079	0.37	1.49	4.31
B3LYP/6-31+G(d,p)	0.876	0.077	0.28	1.50	4.53
B3LYP/6-311+G(d,p)	1.390	0.079	0.36	1.50	4.32
B3LYP-D3/6-311G(d,p)	0.612	0.079	0.37	1.49	4.28
PCM-B3LYP-D3/6-31G(d)	0.369	0.078	0.27	1.50	4.79
PCM-B3LYP-D3/6-31G(d,p)	0.466	0.077	0.27	1.49	4.70
PCM-B3LYP-D3/6-311G(d,p)	0.834	0.078	0.33	1.45	4.16
PCM-B3LYP-D3/6-31+G(d,p)	0.965	0.079	0.25	1.55	4.79
PCM-B3LYP-D3/6-311+G(d,p)	1.570	0.078	0.32	1.49	4.27
ωB97X-D/6-31G(d)	0.414	0.078	0.28	1.52	5.04
ωB97X-D/6-31G(d,p)	0.537	0.080	0.31	1.52	5.00
ωB97X-D/6-311G(d,p)	0.914	0.080	0.35	1.51	4.75
ωB97X-D/6-31+G(d,p)	1.180	0.077	0.25	1.51	4.99
ωB97X-D/6-311+G(d,p)	2.010	0.080	0.34	1.50	4.73
M06-2X/6-31G(d)	0.413	0.079	0.27	1.52	5.13
M06-2X/6-31G(d,p)	0.493	0.078	0.27	1.51	5.09
M06-2X/6-311G(d,p)	0.763	0.081	0.30	1.54	4.91
M06-2X/6-31+G(d,p)	1.095	0.078	0.26	1.51	5.08
M06-2X/6-311+G(d,p)	1.640	0.081	0.30	1.53	4.96
SMD-M06-2X/6-31G(d)	0.685	0.079	0.25	1.52	5.01
SMD-M06-2X/6-31G(d,p)	0.882	0.077	0.26	1.49	4.97
SMD-M06-2X/6-311G(d,p)	1.230	0.079	0.23	1.50	4.80
SMD-M06-2X/6-31+G(d,p)	2.770	0.079	0.25	1.52	4.73
SMD-M06-2X/6-311+G(d,p)	3.600	0.079	0.23	1.50	4.82

^a^ Calculations using PCM (chloroform) solvent model. ^b^ Recommended geometry optimization method highlighted in blue, bold font. ^c^ CPU time for optimization of naupliolide (starting from AM1 geometry). ^d^ RMSD = root-mean-square deviation; MD = maximum deviation.

## Data Availability

Data are available in Electronic Supporting Information (ESI), and for additional details, please contact the authors.

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
