# Peer review of "DELTA50: A Highly Accurate Database of Experimental 1H and 13C NMR Chemical Shifts Applied to DFT Benchmarking"

_molecules, 2023, doi:10.3390/molecules28062449_

Round 1

Reviewer 1 Report

This is to review manuscript molecules-2210237, entitled as “DELTA50: A Highly Accurate Database of Experimental 1H and 13C Chemical Shifts…” by Cohen et al. As its title states, this work is performed to answer the question “which DFT functional reveals better result?”. The authors have selected diverse set of organic species composed of first and second row elements. Although I would prefer systematic analysis, use of diverse sets has its own advantages (not always). In my opinion, this work is probably publishable, subject to some corrections, additional species and analyses:

 1- In general, optimization and analysis must be performed at the same theoretical level, i.e., same method and basis set. Although this rule is based on quantum chemical definitions, in some cases it is violated due to software/hardware/theoretical limitations, normally using double slash (//) notation. However, the fact that “optimization with basis 1 and analysis with basis 2 gives rise to better results” is not sufficient and reasonable to employ distinct bases for optimization and analysis. Therefore, the manuscript should be rewritten in which this flaw is omitted.

2-  I think some small and model molecules are absent, e.g., CH4, H2O, CO, etc. Why?

Suggestion: The considered species are not too large for post-HF methods. I suggest the authors to perform their analyses at MPn, QCI, CC (as much as the authors’ time and computational facilities allow) levels and compare the results.

Author Response

Thank you for taking the time to review the manuscript and for your feedback.

Regarding your first comment, while we completely agree that the geometry optimization and vibrational frequency calculation must be performed using the same functional and basis set, we disagree that this is a requirement for magnetic shieldings and hence chemical shifts.  In fact, it is common practice that this is not the case because geometry optimizations are time consuming versus single point shielding calculations that can be performed at a higher level, particularly with a larger basis set.  This practice has been recommended by many well-respected computational chemists, such as Hoye (Nat Protoc., 2014, 9, 643-660) where PCM-B3LYP/6-311+G(2d,p)//M06-2X/6-31+G(d,p) was employed, as well as the heavily cited and utilized DP4 method of Goodman (J. Am. Chem. Soc., 2010, 132, 12946-12959) that employs fast MMFF geometry optimizations followed by B3LYP/6-31G(d,p) shielding calculations. Furthermore, Table 1 in our manuscript lists the results from 21 previous benchmark studies, and the recommended model chemistries used a different method for geometries and shielding calculations in 20 out of 21 cases.    

Regarding the second comment, while we did include a few very small molecules such as cyclopropane and isobutylene, these compounds are gases at room temperature, and thus it was difficult to control their concentrations in solution, particularly for long 13C acquisitions that were up to 4 hours on a 600 MHz NMR equipped with a sensitive broadband ProdigyTM probe.  They additionally provide few experimental data points to be used for statistical analysis in benchmark studies.  Thus, there is relatively small “bang-for-the-buck” in terms of the necessary effort to measure them.  We do see your point that such small molecules would be of value for post-HF benchmarking, particularly at the level of coupled cluster theory. However, the focus of our paper was on benchmarking DFT methods, which could then assist in structure determination studies of medium-sized to larger organic molecules, including natural products and peptides.  Except MP2, such post-HF calculations would be impractical for these compounds. To emphasize the fact that DELTA50 should be used for DFT benchmarking, we changed the title of the manuscript from “DELTA50:  A Highly Accurate Database of Experimental 1H and 13C Chemical Shifts for Benchmarking Computational Chemistry Methodologies” to “DELTA50: A Highly Accurate Database of Experimental 1H and 13C NMR Chemical Shifts Applied to DFT Benchmarking”  

Reviewer 2 Report

There is a concern regarding the recommendation in Method 1 for geometry optimization. The molecules in the dataset may not have significant issues with the omission of dispersion correction. However, molecules with significant dispersive interactions may experience significant errors in their optimized geometry due to the lack of dispersion correction. This caveat should be clearly stated.

Author Response

Thank you for the feedback regarding the lack of dispersion correction in the geometry optimization step of method 1.  A note was added to the text regarding this point.

Reviewer 3 Report

In this paper, the authors performed a benchmark analysis of 1H and 13C NMR chemical shifts using the DELTA50 library of different organic compounds. The linear scaling factors for conversion of DFT-predicted isotropic magnetic shieldings to chemical shifts was calculated using  75 density functionals, 40 basis sets, 3 solvent models, and 3 gauge referencing schemes. They also evaluated importance of geometry optimisation step.

There are my comments:

1.      Many references are not available, please fix it.

2.      What is this vortex character in Table 1? It looks to me that fonts are messed up.

3.      Figure 3. legend should be formated better, legend text overlaps with legend marks sometimes (for example pyridazine).

The graphic equipment of the article needs to be worked on (images with better resolution, references, the font is changed in several places, strange breaks in the text line), but the content is very good and useful.

Author Response

Thank you for your positive comment regarding the scientific content and for your feedback regarding the typos and graphics.  These were all corrected in the revised manuscript.  Figure 3 was updated for clarity as suggested, and the resolutions of all figures were increased.  Additional references were added to the introduction and throughout the manuscript.

Reviewer 4 Report

The present work may be publishable after thorough editing and taking care of typos. The presence of numerous " (Error! Reference source not found." in the manuscript have made it difficult to follow. The Figure 6 is repeated in the manuscript, so is the paragraph just after the first instance of Figure 6. There are some typos present in the place of symbols and superscripts. If the authors are this much careless about their own work, why do one expect reviewers will want to invest time and efforts to read and understand it.

From the science part I have few comments. 

The authors mentioned "experimental data were not measured in a single solvent system (instead, either CDCl3 or CCl4 was used)." Please clarify. How much difference once would expect in NMR spectra between CDCl3 and CCl4 and why? The dielectric constants are in the very low region. Were their dipole moments responsible for any difference in NMR spectra?

The comparison of the work by Konstantinov et al. vs. that of Toomsalu et al. with regard to the performance of BMK is misleading. Those two studies used different conditions, no solvent vs. toluene continuum. The point should be the effect of different conditions in preparing benchmark data, not on BMK.

"The "fine" integration grid will also improve calculation speed with negligible impact at this level of theory": Please elaborate. The default integration grid in the Gaussian16 code is already ultrafine.  

Author Response

We apologize for the poor quality of the document that was sent out for peer review.  We originally uploaded an extensively checked manuscript, but prior to this being sent out, it was automatically converted into the journal’s template, which introduced the errors. These have all been remedied in the revised manuscript. Additionally, we will use the journal’s template prior to any future first submissions.

We would also like to express our sincere gratitude that you were able to still provide thoughtful and timely comments on the scientific content of our paper. The following are our responses:

  1. Indeed, CDCl3 and CCl4 are both low dielectric solvents but have been shown to yield different NMR spectra. Differences as large as 0.2 ppm for 1H and 6 ppm for 13C chemical shifts between CDCl3 and CCl4 were observed by Stadelmann, et.al. (Phys. Chem. Chem. Phys., 2022, 24, 23551-23560), and a reference to this study was added to our paper.
  2. Thank you for pointing out the method differences between the studies of Konstantinov, et. al. and Toomsalu, et. al. We clarified that different conditions were used per your suggestion and expanded on the topic of conflicting results from various DFT benchmark studies for chemical shift predictions.
  3. Versions prior to Gaussian 16 used the fine integration grid as default, and this typically works well for GGAs and hybrids, such as B3LYP, but may have issues for meta-GGA functionals as discussed by Wheeler and Houk (J. Chem. Theory Comput., 2010, 6, 395-404). Because both method 1 and 2 do not use meta-GGAs, we have found identical NMR predictions when using the fine and ultrafine grids. We further elaborated on the size of the quadrature grid versus performance in our paper.

Round 2

Reviewer 4 Report

Thank you for your response.